# PCP4/PEP19 and HER2 Are Novel Prognostic Markers in Mucoepidermoid Carcinoma of the Salivary Gland

**DOI:** 10.3390/cancers14010054

**Published:** 2021-12-23

**Authors:** Takuya Yoshimura, Shotaro Higashi, Sohsuke Yamada, Hirotsugu Noguchi, Mitsuharu Nomoto, Hajime Suzuki, Takayuki Ishida, Hirotaka Takayama, Yuka Hirano, Masaru Yamashita, Akihide Tanimoto, Norifumi Nakamura

**Affiliations:** 1Department of Oral and Maxillofacial Surgery, Field of Oral and Maxillofacial Rehabilitation, Kagoshima University Graduate School of Medical and Dental Sciences, Kagoshima 890-8520, Japan; higashi@d1.dent.kagoshima-u.ac.jp (S.H.); hajime@dent.kagoshima-u.ac.jp (H.S.); taka-isi@dent.kagoshima-u.ac.jp (T.I.); h-tkym@d1.dent.kagoshima-u.ac.jp (H.T.); y-hrn@d1.dent.kagoshima-u.ac.jp (Y.H.); nakamura@dent.kagoshima-u.ac.jp (N.N.); 2Department of Pathology and Laboratory Medicine, Kanazawa Medical University, Ishikawa 920-0293, Japan; sohsuke@kanazawa-med.ac.jp; 3Department of Pathology, Field of Oncology, Kagoshima University Graduate School of Medical and Dental Sciences, Kagoshima 890-8520, Japan; h-noguchi@med.uoeh-u.ac.jp (H.N.); akit09@m3.kufm.kagoshima-u.ac.jp (A.T.); 4Department of Surgical Pathology, National Hospital Organization Kagoshima Medical Center, Kagoshima 892-0853, Japan; nomoto.mitsuharu.sh@mail.hosp.go.jp; 5Department of Otolaryngology-Head and Neck Surgery, Field of Sensory Organology, Kagoshima University Graduate School of Medical and Dental Sciences, Kagoshima 890-8520, Japan; yamashita@kufm.kagoshima-u.ac.jp

**Keywords:** mucoepidermoid carcinoma, Purkinje cell protein 4/peptide 19, human epidermal growth factor 2, prognosis, salivary gland

## Abstract

**Simple Summary:**

Mucoepidermoid carcinoma (MEC) is the most common malignancy of all salivary neoplasms, and no effective treatment strategy for MEC has been established other than resection. In this study, we showed that Purkinje cell protein (PCP) 4/peptide (PEP) 19 and human epidermal growth factor receptor 2 (HER2) are predicted to play important roles in the pathogenesis and progression of MEC. The detection of PCP4/PEP19 and HER2 may be useful for providing more effective treatments against MEC.

**Abstract:**

Mucoepidermoid carcinoma (MEC) is one of the most common malignant salivary gland carcinomas, but no effective treatment strategy has been established other than surgical resection. Purkinje cell protein (PCP) 4/peptide (PEP) 19 is a calmodulin-binding antiapoptotic peptide that is expressed and inhibits apoptosis in human breast cancer cells. Human epidermal growth factor receptor 2 (HER2) is an epidermal growth factor that has been implicated in the pathogenesis of many carcinomas, particularly breast and gastric carcinomas. In the present study, we performed immunohistochemical analyses of samples from 73 patients who underwent surgical resection for MEC of the salivary gland using antibodies against PCP4/PEP19 and HER2. PCP4/PEP19 expression was related to better prognosis, while HER2 expression was associated with worse prognosis. Patients that were PCP4/PEP19-positive and HER2-negative showed similar outcomes to PCP4/PEP19 and HER2 alone. Therefore, PCP4/PEP19 and HER2 are predicted to play important roles in the pathogenesis and progression of MEC.

## 1. Introduction

Salivary gland carcinomas are a rare and clinically diverse group of neoplasms, among which mucoepidermoid carcinoma (MEC) is the most common malignancy, representing 10 to 15% of all salivary neoplasms [1]. Resection is the standard of care for MEC of the salivary glands, while chemotherapy and radiation are sometimes employed [2]. Gefitinib has been reported to be effective in lung MEC [3], and no effective treatment strategy for MECs of salivary glands has been established other than resection. There are very few alternative treatment modalities available for inoperable cases discovered at an already advanced stage or for cases of postsurgical recurrence, resulting in a poor quality of life and prognosis [4]. The 10-year overall survival rates for low- and intermediate-MECs are approximately 90% and 70%, respectively, but that for high-grade MECs is 25% [5]. Cases of successful molecular-targeted therapy have been reported but have not been established. We previously reported the usefulness of MUC4 and MUC6 in predicting the prognosis of MEC [6], so a classification system according to the molecular expression status may be useful for selecting appropriate therapies, such as molecular-targeted chemotherapy. The majority of salivary gland carcinomas, including MECs, are characterized by recurrent gene fusions, which proved to be highly valuable diagnostically, but less in terms of therapy; for that reason, there is a strong need to find new markers that could also be used in molecular-targeted therapies [7,8].

It has been reported that the expression of human epidermal growth factor receptor 2 (HER2) and epidermal growth factor receptor (EGFR) are poor prognostic factors for several types of cancer [9,10,11]. HER2 and EGFR are known to regulate cell proliferation, differentiation, angiogenesis, and survival [12]. It has been suggested that HER2 may serve as a poor prognostic marker for mucoepidermoid carcinoma of the salivary glands [13].

Purkinje cell protein (PCP) 4/peptide (PEP) 19 was first identified in the rat cerebellum as a 7.6-kDa polypeptide that shows homology to the calcium binding β-chain of the S100 protein [14]. We previously reported PCP4/PEP19 expression in human breast cancer and found that it exerts antiapoptotic functions [15], cell proliferation, invasion, adhesion, and aromatase expression in human breast cancer cells [6,16]. Both breast glands and salivary glands are tubuloacinar exocrine glands sharing similar morphological features; consequently, it is reasonable to expect similarities in their pathological processes [17]. However, no report has investigated the expression of PCP4/PEP19 in salivary gland tumors or MECs to date.

## 2. Materials and Methods

Surgically resected MEC tissues were assessed in the present study. Pathological reports were reviewed to identify patients who underwent simple tumor extirpation or radical sialoadenectomy with cervical lymph node dissection for MEC between 1991 and 2016 at the Unit of Surgical Pathology, Kagoshima University Hospital. A total of 73 patients with available follow-up data were used because the composition was the same as in a previous study [6]. The surgical margins were considered to be involved when MECs at the lateral or deep margin were identified or the distance to the noncarcinomatous mucosa margin was substantially less than 1 mm [18]. The durations of disease-specific survival (DSS) and disease-free survival (DFS) were defined as the interval from the date of surgery to death, except for patients who died from causes other than MEC (DSS) and the interval from the date of surgery to recurrence (DFS). All materials in this article were approved by the Ethical Committee of Kagoshima University Hospital (No. 170101, approved date: 4 August 2017).

The resected tissues were routinely processed for formalin-fixed paraffin-embedded sections and stained with hematoxylin and eosin (H&E) and modified elastic Masson trichrome (EMT) stains. Immunohistochemistry (IHC) was performed using antibodies against podoplanin (D2-40; DAKO, Glostrup, Denmark; diluted 1:1) and S-100 protein (DAKO; diluted 1:20) for the observation of the presence of lymphovascular invasion (LVI) and perineural invasion (PNI), respectively [19].

IHC for PCP4/PEP19, EGFR and HER2 was performed using anti-PCP4 (PEP) (SIGMA, Tokyo, Japan; diluted 1:500), and anti-epidermal growth factor receptor (Leica, Tokyo, Japan; diluted 1:25) antibodies and Hercep Test II for HER2 (kit) (DAKO; diluted ready-to-use). The immunoreactivity was assessed by evaluating the proportion of positive cells among all neoplastic MEC cells. Representative histological images of the expression levels for each protein are shown in Figure 1.

To assess PCP4/PEP19 expression in the cytoplasm and/or the nucleus, areas that had ≥1% positive staining were considered to be positive (Figure 1A). EGFR and HER2 expression were determined based on a previous report [20]; that is, cases where ≥10% of the total neoplastic MECs had a cell membrane that was stained moderately or intensely were regarded as positive (Figure 1B).

The pathological diagnosis was confirmed by three board-certified pathologists (H.N., S.Y., and A.T.) according to the Tumor–Node–Metastasis (TNM) Classification of Malignant Tumors [21]. All MECs were graded based on the three-tiered, low, intermediate, and high histological grading system from the Armed Forces Institute of Pathology (AFIP) proposed by Goode et al. [22]. Clinical information was gathered from the patients’ records. Patients were followed-up and evaluated postoperatively at approximately three- to six-month intervals using physical examination, head, neck, and chest computed tomography (CT) scans and/or measurements of blood cell counts and biochemistry.

All IHC slides were evaluated by two independent observers of board-certified pathologists (H.N. and S.Y.) using a blind protocol design (the observers were blinded to the clinicopathological data). The agreement between the observers was excellent (more than 90%) for all IHC investigated, as measured by the interclass correlation coefficient. For the few (less than 1%) instances of disagreement, a consensus score was determined by a third board-certified pathologist (A.T.) [19].

The significance of correlations was determined using Fisher’s exact test or χ^2^ test, where appropriate, to assess the relationships between IHC and clinicopathological features [19]. Survival curves were plotted with the Kaplan–Meier method and compared by the log-rank test. Hazard ratios and 95% confidence intervals (95% CIs) were estimated using univariate or multivariate Cox proportional hazard regression models [19,23]. All statistical tests were two-tailed, with values of *p* < 0.05 considered to indicate significance. All of the above statistical analyses were performed with Stata 16 (StataCorp LLC, College Station, TX, USA, version 16.1).

## 3. Results

The clinicopathological features of 73 patients with MEC are summarized in Table 1.

The age range at surgery was 12–86 years (average and median were 59.2 and 62 years, respectively). Of the 73 patients, 39 (53.4%) were male and 34 (46.6%) were female. The collected MECs were located at similar rates in the major (*n* = 37, 50.7%; including 30 cases in the parotid gland) and minor (*n* = 36, 49.3%) salivary glands. Tumor stages were defined based on the tumor size at the greatest dimension: T1 (*n* = 24) for ≤2 cm; T2 (*n* = 22) for >2–4 cm; and T3 (*n* = 13) for >4 cm [20]. At diagnosis, 26 patients (35.6%) had lymph node metastases, but no patients (0%) had distant metastases. The tumors included 24 low-grade (32.9%), 12 intermediate-grade (16.4%) and 37 high-grade (50.7%) MECs based on the AFIP criteria [21]. The resection margins of the majority of those MEC specimens (*n* = 55; 75.3%) were free. Postoperative recurrence was noted in 15 of 73 (20.5%) patients. The median DSS and DFS were 37.8 and 42.7 months, respectively (postoperative follow-up data; average: 42.1 months; range: 1–131 months).

Positive expression of PCP4/PEP19, EGFR, and HER2 was recorded in 37 cases (50.7%), 34 cases (46.6%), and 11 cases (15.1%), respectively. The relationship between PCP4/PEP19, EGFR, and HER2 expression and clinicopathological characteristics is summarized in Table 2. PCP4/PEP19 and HER2 expression had a significant correlation with multiple variates (*p* < 0.05), but EGFR expression was not correlated with any variate.

To assess whether PCP4/PEP19, EGFR, and HER2 expression was an independent predictor of postoperative DSS and DFS, Kaplan–Meier curves for DSS and DFS were created (Figure 2).

The results showed that PCP4/PEP19 positivity and HER2 negativity were significantly correlated with prognosis (Figure 2A,C, *p* < 0.05), while no significant correlation was observed between prognosis and EGFR expression (Figure 2B). The combined expression of PCP4/PEP19 and HER2 had a significant correlation with a better prognosis than that with any marker alone (Figure 2D, *p* < 0.05).

Table 3 and Table 4 show the results of the univariate and multivariate Cox proportional hazard regression models with regard to the association between patient prognosis and clinicopathological features.

In the univariate analysis, the following factors were significantly associated with DFS and DSS: N stage, grade, LVI, PNI, necrosis, mitotic figures, positive PCP4/PEP19, and negative HER2. HER2 expression, negative PCP4/PEP19, and positive HER2 status had a significant correlation in DSS. T stage and PCP4/PEP19 expression had a significant correlation with DFS. Positive PCP4/PEP19 resulted in a significantly better prognosis with an HR of less than 1, and the same was true for positive PCP4/PEP19 and negative HER2. The multivariate analysis showed a correlation only between cystic components and DFS.

## 4. Discussion

Here, we investigated the association with PCP4/PEP19, EGFR, and HER2 in MEC, which is the most common malignant salivary gland tumor. To the best of our knowledge, the expression of PCP4/PEP19 in MECs has not been investigated, and in this study, the negative expression rate was 49.3% and the positive expression rate was 50.7%, which was almost 1:1. The expression of EGFR and HER2 has been investigated in the past, and EGFR has been demonstrated to be overexpressed in 25 to 77% of cases, while HER2 is overexpressed in 4.3 to 38% of cases [24,25,26]. EGFR and HER2 were overexpressed in 46.6 and 15.1% of patients in this study, respectively, both of which seem reasonable compared to previous reports. Regarding the statistical relationship between the expression of each protein and clinicopathological factors, PCP4/PEP19 and HER2 expression were found to be correlated with necrosis and mitotic figures as items related to malignant findings by χ^2^ test. Goode et al. [22] scored five variates in the histologic images of MEC and classified the grade into three stages, but in this case, there was no correlation with cystic components, PNI, or anaplasia. The grade showed a correlation only with PCP4/PEP19. El-Attar and Deraz [27] reported that the higher the grade was, the higher the expression level of HER2, but this result suggests that the histology and malignancy of MECs may not correlate with the presence or absence of the expression of PCP4/PEP19 and HER2. The expression of EGFR did not correlate with clinicopathological factors in this study; however, Khiavi et al. [28] reported that there was a statistically significant correlation between EGFR expression and the histopathological grading of MEC of the salivary glands. This study does not rule out a relationship between MEC and EGFR, and further research is needed to clarify this relationship.

Next, in addition to clinicopathological factors, PCP4/PEP19, HER2, and their expression combinations that were correlated with each factor were investigated for their association with prognosis (DSS and DFS). In the univariate analysis, similar to the case of the χ^2^ test above, multiple variates related to the grade had significant correlations with both DSS and DFS, and it has been shown that malignancy can be an important index in predicting prognosis. Furthermore, when PCP4/PEP19 expression was positive, the prognosis was significantly better in terms of DFS, and conversely, when HER2 was positive, the prognosis was significantly worse in terms of DSS. The same was true when PCP4/PEP19 and HER2 were combined, and both PCP4/PEP19-positive and HER2-negative expression were related to a significantly good prognosis in terms of both DSS and DFS; additionally, the log-rank test in Figure 2 also showed that. There have never been reports that the expression of PCP4/PEP19 and HER2 is positively correlated with the prognosis of MEC in the oral region, and our results indicate that the expression of these proteins can be one of the judgment factors in predicting prognosis. The expression of HER2 has been reported in multiple cancers, such as breast cancer and gastric cancer, and the higher the expression is, the worse the prognosis [29,30]. To the best of our knowledge, the relationship between the expression of PCP4/PEP19 and the prognosis of malignant tumors has not yet been reported, but PCP4/PEP19 expression has been reported to promote antiapoptotic effects, migration, invasion, and adhesion of tumor cells in breast cancer [15,16]. Thus, the expression of PCP4/PEP19 is expected to have a negative impact on prognosis in malignant tumors. However, there was no correlation between prognostic factors and immunohistochemical expression in the multivariate analysis, but a correlation was observed in the univariate analysis. The specific reason why no significant difference was observed after controlling for confounders that were important in clinicopathology is unknown, so further research is needed.

There are several limitations associated with the present study. First, we conducted only immunohistochemical analyses and not a detailed molecular analysis, so evaluation at the molecular level could affect the results. Second, the variables considered to be confounding factors were limited, so it is necessary to consider the addition of more detailed factors that have clinicopathological effects in future studies.

## 5. Conclusions

In conclusion, this study demonstrates for the first time that the expression of PCP4/PEP19 and HER2 is associated with the prognosis of MEC, and the results may be useful for treatment planning. These factors are known to be expressed in other areas of cancer; therefore, these findings are expected to facilitate the further development of research.

## Figures and Tables

**Figure 1 cancers-14-00054-f001:**
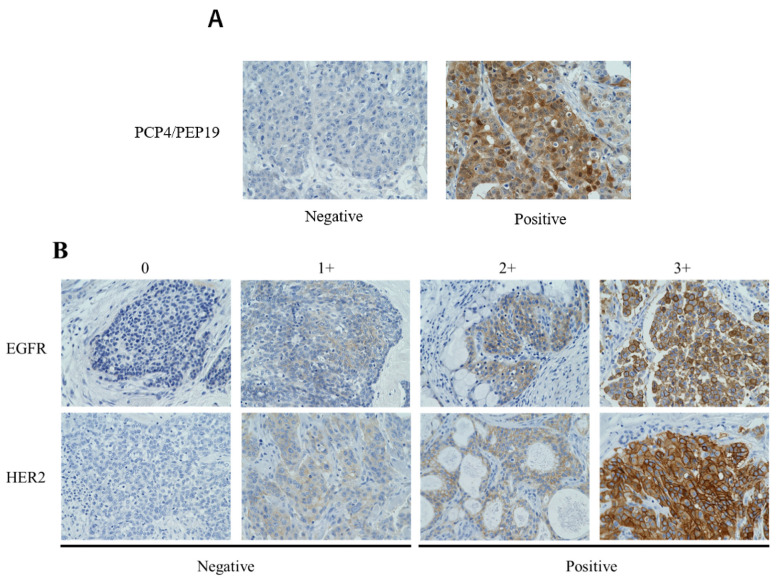
Representative immunohistochemical staining of PCP4/PEP19. (**A**) EGFR and HER2; (**B**) expression in MECs. PCP4/PEP19 expression was judged to be positive if cytoplasmic and nucleic staining was observed in 1% or more of tumor cells. The criteria of positive expression for HER2 were determined by reference to previous reports; that is, the score was divided into four categories (0, 1+, 2+, 3+) according to the membrane staining of tumor cells, and a score of 2+ or more was judged to be positive. Purkinje cell protein 4 (PCP4)/peptide (PEP) 19, brain-specific polypeptide; EGFR, epidermal growth factor receptor; HER2, human epidermal growth factor receptor type 2; MEC, mucoepidermoid carcinoma.

**Figure 2 cancers-14-00054-f002:**
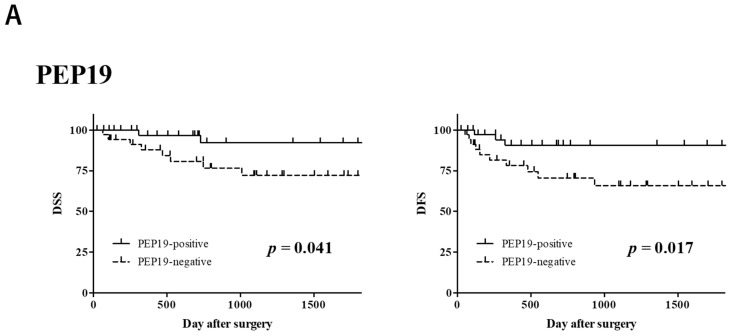
Kaplan–Meier curves for the DFS and DSS of MEC patients within the first five years after surgery, according to PCP4/PEP19 (**A**); EGFR (**B**); and HER2 (**C**) expression. Since PCP4/PEP19 and HER2 were significantly related to DSS and DFS, the Kaplan–Meier curve is shown for combined expression (**D**). DFS, disease-free survival; DSS, disease-specific survival; MEC, mucoepidermoid carcinoma; Purkinje cell protein 4 (PCP4)/peptide (PEP) 19, brain-specific polypeptide; EGFR, epidermal growth factor receptor; HER2, human epidermal growth factor receptor type 2; PEP19(+)HER2(+), PCP4/PEP19-positive and HER2-positive; PEP19(+)HER2(−), PCP4/PEP19-positive and HER2-negative; PEP19(−)HER2(+), PCP4/PEP19-negative and HER2-positive; PEP19(−)HER2(−), PCP4/PEP19-negative and HER2-negative.

**Table 1 cancers-14-00054-t001:** Clinicopathological variables of all MEC patients.

Characteristic	Patients (*n* = 73)	Characteristic	Patients (*n* = 73)
Age(years)		PNI	
Average	59.2	(−)	44
Median	62	(+)	29
Range	12–86	Necrosis	
>60	40	(−)	31
≤60	33	(+)	42
Sex		Anaplasia	
Male	39	(−)	66
Female	34	(+)	7
T stage		Mitotic figures	
T1	24	<4	35
T2	22	4≤	38
T3	13	Margin status	
T4	14	(−)	55
N stage		(+)	18
N0	47	Months after surgery	
N1	6	Average	42.1
N2	20	Median	33
TNM stage		Range	1–131
Stage I, II	32	Recurrence	
Stage III, IV	41	(−)	58
Grade		(+)	15
Low	24	Location	
Intermediate	12	Major salivary gland	37
High	37	Parotid gland	30
Cystic components		Submandibular gland	4
<20%	55	Sublingual gland	3
≥20%	18	Minor salivary gland	36
LVI		Palatinal gland	6
(−)	28	Other minor gland	30
(+)	45		

LVI, lymphovascular invasion; PNI, perineural invasion.

**Table 2 cancers-14-00054-t002:** The relationship between PCP4/PEP19 expression and EGFR/HER2 expression.

Variables		PCP4/PEP19 Expression			EGFR Expression			HER2 Expression		
		Negative (*n* = 36, 49.3%)	Positive (*n* = 37, 50.7%)	*p* Value	Negative (*n* = 39, 53.4%)	Positive (*n* = 34, 46.6%)	*p* Value	Negative (*n* = 62, 84.9%)	Positive (*n* = 11, 15.1%)	*p* Value
Age										
>60	40 (54.8)	15	18	0.64	18	22	0.158	33	7	0.744
≤60	33 (45.2)	21	19		21	12		29	4	
Sex										
Male	39 (53.4)	21	18	0.484	18	21	0.241	34	5	0.745
Female	34 (46.6)	15	19		21	13		28	6	
Location										
Major salivary gland	37 (50.7)	12	25	**0.005**	21	16	0.642	30	7	0.515
Minor salivary gland	36 (49.3)	24	12		18	18		32	4	
T stage										
T1, 2	46 (63.0)	23	23	1	26	20	0.628	41	5	0.309
T3, 4	27 (37.0)	13	14		13	14		21	6	
N stage										
N0	47 (64.4)	21	26	0.334	29	18	0.086	43	4	**0.046**
N1, 2	26 (35.6)	15	11		10	16		19	7	
Grade										
Low, Intermediate	36 (49.3)	11	25	**0.002**	20	16	0.816	33	3	0.19
High	37 (50.7)	25	12		19	18		29	8	
Cystic components										
<20%	55 (75.3)	28	27	0.787	10	8	1	17	1	0.273
≥20	18 (24.7)	8	10		29	26		45	10	
LVI										
(−)	28 (38.4)	12	16	0.472	17	11	0.347	27	1	**0.042**
(+)	45 (61.6)	24	21		22	23		35	10	
PNI										
(−)	44 (60.3)	21	23	0.813	25	19	0.632	40	4	0.101
(+)	29 (39.7)	15	14		14	15		22	7	
Necrosis										
(−)	31 (42.5)	10	21	**0.018**	19	12	0.343	30	1	**0.019**
(+)	42 (57.5)	26	16		20	22		32	10	
Anaplasia										
(−)	66 (90.4)	32	34	0.711	35	31	1	57	9	0.283
(+)	7 (9.6)	4	3		4	3		5	2	
Mitotic figures										
<4	35 (47.9)	11	24	**0.005**	20	15	0.64	34	1	**0.007**
≥4	38 (52.1)	25	13		19	19		28	10	
Margin status										
(−)	55 (75.3)	28	27	0.787	27	28	0.277	50	5	**0.022**
(+)	18 (24.7)	8	10		12	6		12	6	
Recurrence										
(−)	58 (79.5)	25	33	**0.046**	33	25	0.263	51	7	0.221
(+)	18 (20.5)	11	4		6	9		11	4	
Tumor-related death										
(−)	63 (86.3)	28	35	**0.046**	36	27	0.172	56	7	**0.038**
(+)	10 (13.7)	8	2		3	7		6	4	

Purkinje cell protein 4 (PCP4)/peptide (PEP) 19, brain-specific polypeptide; EGFR, epidermal growth factor receptor; HER2, human epidermal growth factor receptor type 2; LVI, lymphovascular invasion; PNI, perineural invasion.

**Table 3 cancers-14-00054-t003:** Univariate Cox proportional hazards analysis of clinicopathological variates and predictors of DSS and DFS.

Variables	*n* (%)	Univariate					
		DSS			DFS		
		HR	95% CI	*p* Value	HR	95% CI	*p* Value
Age							
>60	40(54.8)	1			1		
≤60	33(45.2)	0.44	0.11–1.72	0.24	0.47	0.16–1.38	0.17
Sex							
Male	39(53.4)	1			1		
Female	34(46.6)	0.67	0.19–2.38	0.54	0.59	0.20–1.67	0.32
Location							
Major salivary gland	37(50.7)	1			1		
Minor salivary gland	36(49.3)	2.4	0.62–9.27	0.21	1.51	0.54–4.26	0.43
T stage							
T1, 2	46(63.0)	1			1		
T3, 4	27(37.0)	2.82	0.79–10.01	0.11	4.18	1.43–12.26	**0.009**
N stage							
N0	47(64.4)	1			1		
N1, 2	26(35.6)	5.05	1.30–19.57	**0.02**	7.52	2.37–23.87	**0.001**
Grade							
Low, intermediate	36(49.3)	1			1		
High	37(50.7)	11.45	1.45–90.62	**0.02**	11.49	2.51–52.60	**0.002**
Cystic components							
<20%	55(75.3)	1			1		
20≤	18(24.7)	0.28	0.36–2.25	0.23	0.32	0.072–1.45	0.14
LVI							
(−)	28(38.4)	1			1		
(+)	45(61.6)	9.52	1.23–73.38	**0.031**	12.62	1.65–96.67	**0.015**
PNI							
(−)	44(60.3)	1			1		
(+)	29(39.7)	4.11	1.06–15.97	**0.04**	4.37	1.46–13.14	**0.009**
Necrosis							
(−)	31(42.5)	1			1		
(+)	42(57.5)	14.23	1.86–108.71	**0.011**	18.50	2.39–143.29	**0.005**
Anaplasia							
(−)	66(90.4)	1			1		
(+)	7(9.6)	1.03	0.13–8.13	0.98	3.49	0.95–12.73	0.059
Mitotic figures							
<4	35(47.9)	1			1		
≥4	38(52.1)	17.19	2.25–131.49	**0.006**	21.50	2.79–165.51	**0.003**
Margin status							
(−)	55(75.3)	1			1		
(+)	18(24.7)	1.96	0.55–6.96	0.3	1.54	0.53–4.51	0.43
PCP4/PEP19 expression							
negative	36(49.3)	1			1		
positive	37(50.7)	0.23	0.048–1.08	0.063	0.27	0.085–0.86	**0.027**
HER2 expression							
negative	62(84.9)	1			1		
positive	11(15.1)	4.93	1.367–17.81	**0.015**	3.23	0.99–10.58	0.053
PCP4/PEP19(+)HER2(+)							
N/A	66(90.4)	1			1		
applicable	7(9.6)	3.30	0.68–15.89	0.137	2.15	0.47–9.76	0.321
PCP4/PEP19(+)HER2(−)							
N/A	43(58.9)	1			1		
applicable	30(41.1)	0.15	0.034–0.69	**0.014**	0.14	0.032–0.65	**0.012**
PCP4/PEP19(−)HER2(+)							
N/A	69(94.5)	1			1		
applicable	4(5.5)	4.749	1.00–22.46	**0.049**	4.00	0.88–18.20	0.073
PCP4/PEP19(−)HER2(−)							
N/A	41(56.2)	1			1		
applicable	32(43.8)	2.003	0.565–7.102	0.282	2.32	0.82–6.57	0.11

HR, hazard ratio; CI, confidence interval; DSS, disease-specific survival; DFS, disease-free survival; Purkinje cell protein 4 (PCP4)/peptide (PEP) 19, brain-specific polypeptide; EGFR, epidermal growth factor receptor; HER2, human epidermal growth factor receptor type 2; LVI, lymphovascular invasion; PNI, perineural invasion; PCP4/PEP19(+) HER2(+), PCP4/PEP19-positive and HER2-positive; PCP4/PEP19(+)HER2(−), PCP4/PEP19-positive and HER2-negative; PCP4/PEP19(−)HER2(+), PCP4/PEP19-negative and HER2-positive; PCP4/PEP19(−)HER2(−), PCP4/PEP19-negative and HER2-negative; N/A, not applicable, PCP4/PEP19 and/or HER2 expression are not applicable.; applicable, PCP4/PEP19 and HER2 expression are applicable.

**Table 4 cancers-14-00054-t004:** Multivariate Cox proportional hazards analysis of clinicopathological variates and predictors of DSS and DFS.

Variables	*n* (%)	Multivariate					
		DSS			DFS		
		HR	95% CI	*p* Value	HR	95% CI	*p* Value
Age							
>60	40(54.8)	1			1		
≤60	33(45.2)	0.38	0.055–2.59	0.321	0.57	0.11–2.88	0.495
Sex							
Male	39(53.4)	1			1		
Female	34(46.6)	6.47	0.35–119.61	0.210	3.04	0.43–21.25	0.263
Location							
Major salivary gland	37(50.7)	1			1		
Minor salivary gland	36(49.3)	1.08	0.13–9.00	0.943	0.46	0.60–3.51	0.454
T stage							
T1, 2	46(63.0)	1			1		
T3, 4	27(37.0)	4.45	0.41–47.64	0.218	4.68	0.97–22.68	0.055
N stage							
N0	47(64.4)	1			1		
N1, 2	26(35.6)	1.20	0.14–10.17	0.868	5.33	0.66–43.01	0.116
Grade							
Low, intermediate	36(49.3)	1			1		
High	37(50.7)	0.071	0.00078–6.40	0.249	0.61	0.029–13.04	0.753
Cystic components							
<20%	55(75.3)	1			1		
20≤	18(24.7)	34.94	0.70–1756.31	0.075	103.25	2.97–3587.47	**0.010**
LVI							
(−)	28(38.4)	1			1		
(+)	45(61.6)	4.17	0.0028–6297.00	0.702	3.62	0.012–1061.23	0.657
PNI							
(−)	44(60.3)	1			1		
(+)	29(39.7)	0.80	0.080–8.01	0.850	0.39	0.050–3.06	0.370
Necrosis							
(−)	31(42.5)	1			1		
(+)	42(57.5)	20.71	0.0036–119547.5	0.493	10.78	0.0015–75416.37	0.599
Anaplasia							
(−)	66(90.4)	1			1		
(+)	7(9.6)	0.083	0.0044–1.58	0.098	0.72	0.11–4.81	0.738
Mitotic figures							
<4	35(47.9)	1			1		
≥4	38(52.1)	54.88	0.010–300795.8	0.362	23.81	0.0046–122684.8	0.467
Margin status							
(−)	55(75.3)	1			1		
(+)	18(24.7)	10.77	0.42–273.47	0.150	3.95	0.46–34.20	0.212
PCP4/PEP19 expression							
negative	36(49.3)	1			1		
positive	37(50.7)	0.18	0.0060–5.42	0.325	0.83	0.050–13.61	0.894
HER2 expression							
negative	62(84.9)	1			1		
positive	11(15.1)	0.43	0.014–12.68	0.624	2.02	0.074–55.44	0.677
PCP4/PEP19(+)HER2(+)							
N/A	66(90.4)	1			1		
applicable	7(9.6)	15.68	0.94–261.27	0.055	1.05	0.087–12.55	0.972
PCP4/PEP19(+)HER2(−)							
N/A	43(58.9)	1			1		
applicable	30(41.1)	0.31	0.040–2.40	0.261	0.22	0.029–1.63	0.137
PCP4/PEP19(−)HER2(+)							
N/A	69(94.5)	1			1		
applicable	4(5.5)	1.08	0.12–9.56	0.947	0.94	0.11–7.82	0.951
PCP4/PEP19(−)HER2(−)							
N/A	41(56.2)	1			1		
applicable	32(43.8)	1.78	0.21–15.19	0.600	5.12	0.71–36.79	0.104

HR, hazard ratio; CI, confidence interval; DSS, disease-specific survival; DFS, disease-free survival; Purkinje cell protein 4 (PCP4)/peptide (PEP) 19, brain-specific polypeptide; EGFR, epidermal growth factor receptor; HER2, human epidermal growth factor receptor type 2; LVI, lymphovascular invasion; PNI, perineural invasion; PCP4/PEP19(+)HER2(+), PCP4/PEP19-positive and HER2-positive; PCP4/PEP19(+)HER2(−), PCP4/PEP19-positive and HER2-negative; PCP4/PEP19(−)HER2(+), PCP4/PEP19-negative and HER2-positive; PCP4/PEP19(−)HER2(−), PCP4/PEP19-negative and HER2-negative; N/A, not applicable, PCP4/PEP19 and/or HER2 expression are not applicable.; applicable, PCP4/PEP19 and HER2 expression are applicable.

## Data Availability

Data are available on request due to restrictions of ethical policy.

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
