# Peer review of "PCP4/PEP19 and HER2 Are Novel Prognostic Markers in Mucoepidermoid Carcinoma of the Salivary Gland"

_cancers, 2021, doi:10.3390/cancers14010054_

Round 1
Reviewer 1 Report
Brief summary: This article analyzes PEP19, EGFR, and HER2 immunohistochemical expression in 73 patients with salivary mucoepidermoid. This provides fascinating insight into possible predictors of disease that would be helpful in appropriately counseling patients prior to surgery.
Strengths: The novelty of analyzing the expression of PEP19 is a great addition to the literature. This manuscript is clear and relevant to the audience of Cancers. The scientific methodology is clear and statistical analyses reasonable. Figures are clear, helpful, and appropriate. The authors’ conclusions are reasonable.
Weaknesses: Unclear how well visualized expression form IHC compares with protein expression via Western blot. Little mention of other treatment options and other manuscripts that investigate HER2 expression in these tumors.
There are some other treatments for MEC of the salivary glands that it would be helpful to mention, while emphasizing that resection – while not the only option - is the standard of care. It would be good to mention and cite that chemotherapy and radiation are sometimes employed. (https://pubmed.ncbi.nlm.nih.gov/27040585/)
The author’s comparison of PEP positive and HER2 negative in contrast to “single-positive” is confusing and not clear on line 42.
Line 64-65. HER2 has been previously investigated in salivary MEC for its prognostic capacities and this should be mentioned and referenced appropriately (https://pubmed.ncbi.nlm.nih.gov/14974865/ among others).
Reviewer 2 Report
The paper „PEP19 and HER2 are Novel and Reliable Markers for Prognosis in Mucoepidermoid Carcinoma of the Salivary Gland”, although rather modest in terms of sample size and number of analyzed parameters, is relevant to the field of head and neck cancer biology. Even more, it has a practical value as it offers new potential biomarkers/predictors of mucoepidermoid carcinomas (MEC) outcome. Studies on this topic are scarce, yet very much needed. The paper deals with immunohistochemical expression of PCP4/PEP19, EGFR and HER2 in 73 samples of MEC. Authors established several statistically significant associations between PEP and HER expression and clinico-pathological parameters of MEC, the main findings being that PCP4/PEP19 expression was related to better prognosis while HER2 expression was associated with poor prognosis. The paper is interesting and deserves to be published, but I have however a few comments/criticisms:
TITLE: I suggest a slight change of the title “PCP4/PEP19 and HER2 are Novel Prognostic Markers in Mucoepidermoid Carcinoma of the Salivary Gland”. Reliability must be confirmed by other studies.
(Use the complete abbreviation PCP4/PEP19 in the title and throughout the manuscript).
INTRODUCTION:
You should mention in one or two sentences that the majority of salivary gland carcinomas, including MECs, are characterized by recurrent gene fusions, which proved to be highly valuable diagnostically, but less in terms of therapy, and for that reason there is a strong need of finding new markers that could be also used in molecular-targeted therapies. Cite DOI: 10.1172/jci.insight.139497 and DOI: 10.2298/GENSR1402601D
Lines 50-51: rephrase, because it is unclear “Gefitinib has been reported to be effective in lung MEC, and no effective treatment strategy for MEC has been established other than resection”. I guess that authors meant “...for MECs of salivary glands has been established...”.
Lines 56-57: again a sentence that is not clear and that must be rephrased “Cases of successful molecular-targeted therapy have been reported but have not been established”. What do you mean by “have been reported but not established”?
Line 63: you definitely need some citations for the statement that HER2 has been related to different types of cancers:
Oral cancer DOI: 10.1016/j.ijom.2018.01.020
Gastroesophageal cancer: DOI: 10.1002/cac2.12214
Endometrial cancer: DOI: 10.5306/wjco.v12.i10.868
(you can also leave the references 21 and 22 that you gave in the Discussion section)
M&M
Line 83: „less than 1mm“, do you have a reference for this?
Do you have a negative control for immunohistochemistry (omitting the antibodies)?
RESULTS
Line 150: the title of Table 1 is incorrect; you have only clinico-pathological data here.
In their Tables, authors should highlight the values that are statistically significant, either by bolding them or by putting asterisks, because it is not easy to see them.
Table 3: you should explain what do you mean by “applicable” and “not applicable”.
Univariate and multivariate analyses should be separated and make two different tables.
Line 190: when you comment Table 3, you could also mention the parameters with protective effect, i.e, with HR<1.
DISCUSSION
Lines 201-205: rephrase, do not repeat twice that you analyzed PEP19, EGFR, and HER2.
English language/style should be improved.
